# Analysis of Contact Force and Shape Change on Grasping a Square Object Using an Actual Fin Ray Soft Gripper

**DOI:** 10.3390/s23249827

**Published:** 2023-12-14

**Authors:** Takahide Kitamura, Kojiro Matsushita, Naoki Nakatani

**Affiliations:** Department of Mechanical Engineering, Gifu University, Gifu 501-1193, Japan; matsushita.kojiro.h7@f.gifu-u.ac.jp (K.M.); nakatani.naoki.i8@s.gifu-u.ac.jp (N.N.)

**Keywords:** soft robotics, robotics, soft robot gripper, Fin Ray effect, gripping performance evaluation

## Abstract

The Fin Ray-type soft gripper (FRSG) is a typical soft gripper structure and applies the deformation characteristics of the Fin Ray structure. This structure functions to stabilize the grasping of an object by passive deformation due to external forces. To analyze the performance of detailed force without compromising the actual FRSG characteristics, it is effective to incorporate multiple force sensors into the grasping object without installing them inside the Fin Ray structure. Since the grasping characteristics of the FRSG are greatly affected by the arrangement of the crossbeams, it is also important to understand the correspondence between the forces and the geometry. In addition, the grasping characteristics of an angular object have not been verified in actual equipment. Therefore, in this study, a contact force measurement device with 16 force sensors built into the grasping object and a structural deformation measurement device using camera images were used to analyze the correspondence between force and structural deformation on an actual FRSG. In the experiment, we analyzed the influence of the crossbeam arrangement on the grasping force and the grasping conditions of the square (0°) and rectangular (45°) shapes, and state that an ideal grasp in a square-shaped (45°) grasp is possible if each crossbeam in the FRSG is arranged at a different angle.

## 1. Introduction

In recent years, robots that perform tasks in place of humans, such as cooperative robots, have been attracting increasing attention in the field of high-diversity, low-volume production. Research and development have been progressing accordingly. One of the important functions of such robots is grasping, and gripper research and development has been actively conducted. Major gripper configurations are rigid-link, multi-degree-of-freedom motor structures that realize stable grasping of a wide variety of objects through precise angle control based on sensor feedback. However, such grippers are expensive to control and require time-consuming mounting work each time the gripping object is changed. In contrast, with the development of 3D model design and 3D printer fabrication, soft grippers that realize stable grasping with simple control, assuming flexible materials and motor structures with low DOF, are attracting attention [1]. There are many soft grippers with various flexible structures, such as Soft Robotics’ mGrip [2] and Empire Robotics’ VERSABALL^®^ Gripper [3]. Soft grippers can be classified into two main types. One is the pneumatically driven soft gripper, which allows compliant wrapping and a grasping motion due to the compressibility and fluidity of air; Dilibal et al. [4] and Sun et al. [5] have proposed this type of soft gripper. The second is a soft gripper that mimics a biological system. Soft grippers, which mimic biological systems, are known for their unique flexible structure and functionality to realize stable grasping. These functions include the ability to passively deform with the force generated by contact with an object and to adapt to the shape of the object [6]. Such functional structures not only reduce the control burden, but also enable the stable grasping of objects of different sizes within a certain range with the same soft gripper. In addition, it has been found that changing the angle and thickness of the structure of a specific part further contributes to the stable gripping of objects in a wide range of different shapes. These advantages are thought to reduce the time cost of frequent gripper mounting operations in a high-mix, low-volume production, and the gripper is attracting attention as an effective gripper for recent production sites. A representative example is the Fin-Ray-type soft gripper (FRSG), which uses the Fin Ray^®^ effect [7] proposed by Leif Kniese et al. in 1997. The Fin Ray^®^ effect is a bio-mimetic structural mechanism (Fin Ray structure). It is based on the structure of fish fins, and consists of a front beam and back beam, which are the oblique sides of a triangle, and a crossbeam connecting them. When the front beam receives a contact force from the grasping object, the internal force is transmitted to the crossbeam, and the entire structure is passively deformed by wrapping in the direction of the grasping object. In other words, the passive grasping motion is realized by using two Fin Ray structures to clamp the grasping object. The Fin Ray structure has been released as an open-sourced design [8,9], and many researchers and developers have customized the Fin Ray structure according to the target grasping object. For example, Hemming et al. [10] developed a robot for harvesting agricultural crops such as green peppers, and realized the grasping of various types of peppers with different sizes and shapes. However, to introduce the most efficient FRSG for a production site, it is necessary to analyze whether the structure is effective for the grasping object. For this reason, finite element method (FEM) analysis has been the mainstream method used in conventional research, especially in attempts to verify the grasping performance when the crossbeam structure is changed. For example, Elgeneidy et al. [11] reported the ability to change the stiffness of the Fin Ray structure due to contact between crossbeams by changing the angle and arrangement of the crossbeams. Manoonpong et al. [12] used a Fin Ray structure for the feet of a multi-legged mobile robot and verified the angle at which it becomes stable on each uneven terrain by changing the crossbeam angle. This FEM has shown some degree of success in terms of demonstrating the grasping performance of the FRSG model in the simulation. However, the material properties and geometry of the actual 3D printed objects may not always be in perfect agreement with the FEM analysis model, and it is difficult to achieve perfect agreement in the grasping behavior. In addition, it is difficult to analyze buckling and slippage by FEM simulation. Therefore, we recognize that it is important to produce an actual FRSG using a 3D printer, in order to carry out analysis and evaluation, instead of using a simulation. In the grasping state analysis evaluation of an FRSG, it is important to analyze the relationship between the force and shape of the Fin Ray structure. There have been four types of evaluation methods used in previous studies using actual equipment. The first method involves building sensors into the Fin Ray structure: Zhou et al. [13] used piezoelectric elements, Zapciu et al. [14] used capacitance sensors inside the Fin Ray structure, and Hashizume et al. [15] used Gecko sheets and taxel sensor sheets inside the FRSG to analyze the force at each structural position. The second method is to place a flexible structure with feature points inside the Fin Ray structure, record its deformation with a camera, and use image processing to estimate the contact force state from the amount of deformation. Sandra et al. [16] placed a gel sheet with feature points and a small camera inside the Fin Ray structure and used image processing to estimate the contact shape and force by linearly deforming the gel sheet. Furthermore, GelSight Baby Fin Ray [17,18], which minimizes the sensor-unrecognizable area by devising the camera arrangement, has also been developed. However, these two methods are considered to affect the grasping characteristics of the FRSG, and the part with a built-in sensor is easily damaged. Furthermore, when multiple sensors are arranged, there is a problem in that the part that cannot be measured is generated. The third type is a method in which the force sensor is built into the grasping object. This method does not interfere with the grasping motion of the FRSG, and the sensor position and contact area can be changed, enabling flexible operation. Furthermore, it enables more accurate measurement and analysis than capacitive or piezoelectric methods. For example, Shan et al. [19] and Suder et al. [20] have analyzed 6-axis force applied to the center of a cylindrical object by dividing the object into two parts and installing a 6-axis force sensor in the center joint of the part. However, because this method analyzes the composite force applied to one central point, it has not yet demonstrated the wrapping and grasping characteristics of the FRSG with quantitative data. To quantitatively evaluate the wrapping grasp, analysis using a measurement system with high spatial resolution is necessary. The fourth method type is one in which the elastic properties of each part of the Fin Ray structure are known in advance, the amount of deformation is measured from camera image analysis, and force estimation is performed based on the relationship between the elastic properties of the specific part and the amount of deformation. Stuhne et al. [21] succeeded in estimating the internal force from camera image analysis of the side profile of a Fin Ray structure. However, this method is technically difficult because it requires a high-precision understanding of the elastic properties of the Fin Ray structure to achieve a highly accurate estimation. In contrast, Barrie et al. [22] and Deng et al. [23] constructed an elastic property model of the Fin Ray structure based on the actual device in an FEM simulation. This method numerically analyzes grasping with an object model under various conditions and machine learning is employed for the relationship between force and shape. Then, the shape of the object grasped by the actual FRSG is input to the machine learning system, and an operationally efficient system for estimating the force is realized. However, this FEM–machine learning method is currently applicable only under the precondition that stable grasping is possible with little buckling or slippage. In other words, there is no problem with cylinders, which are the basic object of the FRSG, but the grasping of objects with angles, such as a square object, results in a different behavior. This is because the shape and material properties of the FRSG and the object have a large influence on the behavior of the object, and it is important to evaluate the actual object instead of FEM analysis. Based on the above four previous studies, we believe that it is essential to establish an evaluation process for FRSGs to achieve efficient operation at high-mix, low-volume production sites. To realize such an evaluation process, it is necessary to use a system that combines a high-spatial-resolution force measurement device with multiple force sensors built into the gripping object and a camera image analyzer for measuring FRSG side deformation. In the previous study, we developed a 16-way object contact force measurement system and a structural deformation measurement system based on camera image analysis and analyzed quantitative data from a case of grasping a cylinder, which is the basic grasping object, using the standard FRSG open-sourced by Festo. The results show that the Fin Ray structure deforms passively and successfully exhibits a distributed contact force state that indicates an enveloping grasp on the object [24].

As the next step in this study, we will use these systems to analyze five types of enveloping grasping characteristics with different crossbeam angles, which are used in many applications of FRSGs. Furthermore, we will attempt to analyze not only cylinders, which are basic grasping objects, but also square objects with angles (two different arrangements), which are difficult to analyze properly with FEM simulations.

## 2. Proposed System

This chapter describes the configuration of built-in and contact force measurement systems for two types of grasping object for the proposed Fin-Ray-type soft gripper (FRSG) and a camera-image-based structural displacement measurement system.

### 2.1. Proposed Method for Analyzing the Wrapping and Grasping Characteristics of FRSGs

Figure 1 shows an external view of the contact force measurement system of the 16/16 type used in the experiment. Type 16/16 divides the circumference into 16 segments (22.5 degrees), and a single-axis force sensor is attached to each segmented arc. The measurement axis direction of the single-axis force sensor is parallel to the line connecting a point on the circumference and the center of the cylinder (normal direction). This enables batch measurement of the contact force in the direction normal to the FRSG applied to the surface of the 16 segmented objects of the cylinder. Each single-axis force sensor has a cut-out aluminum plate part (Figure 1b) fixed to it. As is shown in Figure 1c, the sockets can be either cylindrical or rectangular in shape. In the experiments, it is possible to use the cylindrical shape, with the square shape set at 0° to the FRSG, and the square shape set at 45° to the FRSG. As is shown in Figure 2, the bottom of the contact force measurement system is fixed to the aluminum frame by a dedicated aluminum plate. The fixing method between the dedicated fixing aluminum plate and the aluminum frame can be switched between simple screw fixing (forward/backward movement: fixed) and fixing via a slider mechanism (forward/backward movement: non-fixed). When using a slider mechanism for fixation, the amount of forward/backward movement of the cylinder can be measured with an accuracy of 200 µm using a Panasonic micro laser rangefinder (HG-C1100, Panasonic Corporation, Tokyo, Japan). Kent paper was attached to the side of the aluminum plate dedicated for fixing. By irradiating the laser beam onto the Kent paper, the reflected laser beam was stabilized. On the other hand, for data measurement analysis of the 16-segment contact force distribution, 16-channel analog voltage outputs were acquired by connecting the instrumentation amplifier circuit to a single-axis force sensor. (load cell: rated capacity 2 kg). Analog signal data were recorded on a PC: PC1 (sampling rate: 1 kHz, resolution: 16 bit) using data acquisition: DAQ (National Instruments, USB-6218, Asahi Engineering Co. Ltd., Tokyo, Japan). (Figure 1a) The data measured via the DAQ device were batch processed in the software MATLAB (Version: 9.14.0.2286388, R2023a) on the same PC1. The flow of analysis consisted of (1) reading the time series data of 10 measurements, (2) calculating the average of 10 measurements, (3) removing high-frequency noise by using a Savitzky–Golay filter on the average, and (4) displaying the results in a graph.

### 2.2. The FRSG Structural Deformation Measurement System by Camera Image Analysis

The hardware configuration of the measurement system for measuring the structural deformation of the FRSG is shown in Figure 2. The system consists of a USB camera (ELP 4 K from AilipuTechnolog Co, Ltd., Guangdong, China), a USB camera position control motor (Dynamixel XM540 from Robotis Co, Ltd., Seoul, Republic of Korea), a group of color markers installed on the side of the FRSG, and a computer (PC2) for the measurement and analysis of camera images. As is shown in Figure 3 and Figure 4, a total of 23 red markers were installed on one finger side of the FRSG. The markers were located on the intersection of the main beam and the crossbeam. The camera was placed above the Fin Ray structure with markers so that all markers could be recorded in the camera’s angle of view. The camera base could also be moved by motor control so that this USB camera could follow this motion of the Fin Ray structure with markers. This ensures that the lower part of the finger of the FRSG with a marker is always in a fixed position in the image. The grasping motion was recorded in video (1280 × 720 px, 60 fps) on PC2. The software of the structural displacement analysis system used for image measurement and analysis was built on the measurement and analysis software MATLAB with the Image Processing Toolbox. The flow of video processing was as follows: (1) convert video files to a group of still image files, (2) extract only the red marker portion of all still image files by color recognition, (3) binarize the red portion (marker), (4) assign a number label (No.1 to No.23) to each marker using a Kalman filter, and (5) calculate (6). Finally, a digital signal filter (Savitzky–Golay filter) was applied to remove high-frequency noise and obtain smoothed displacement data. This allowed us to analyze the displacement of each node relative to a straight line connecting the front and back beam root markers of the FRSG [25,26].

**Figure 2 sensors-23-09827-f002:**
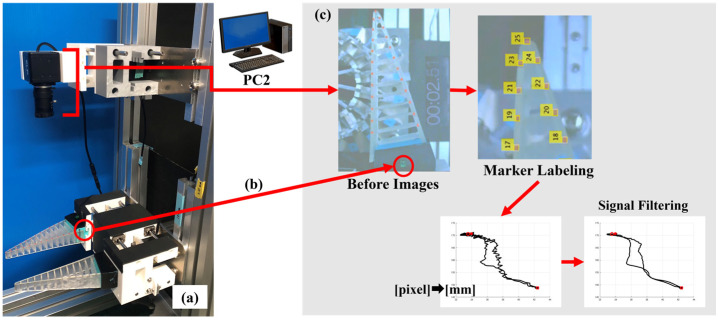
Structural deformation measurement system using camera image processing. (**a**) Appearance of the device; (**b**) LED for synchronization signal between camera image and contact force data; (**c**) Flow of analysis of acquired image data (The red dots represent the point of motion and the point of stable grasp, respectively.).

**Figure 3 sensors-23-09827-f003:**
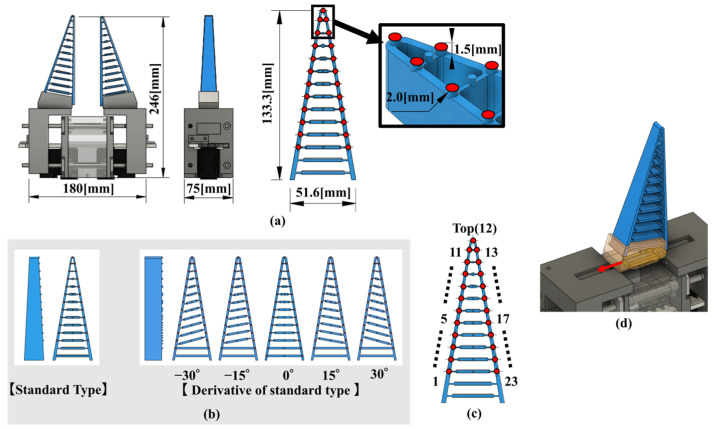
Specifications of the FRSG capable of parallel open/close motion. (**a**) Dimensions and outline of the FRSG and size of the markers; (**b**) Five different Fin Ray structures used in the experiment; (**c**) Number labels for each marker; (**d**) Fin Ray structure fixing mechanism with T-slot and a finger can be inserted in the direction of the red arrow.

**Figure 4 sensors-23-09827-f004:**
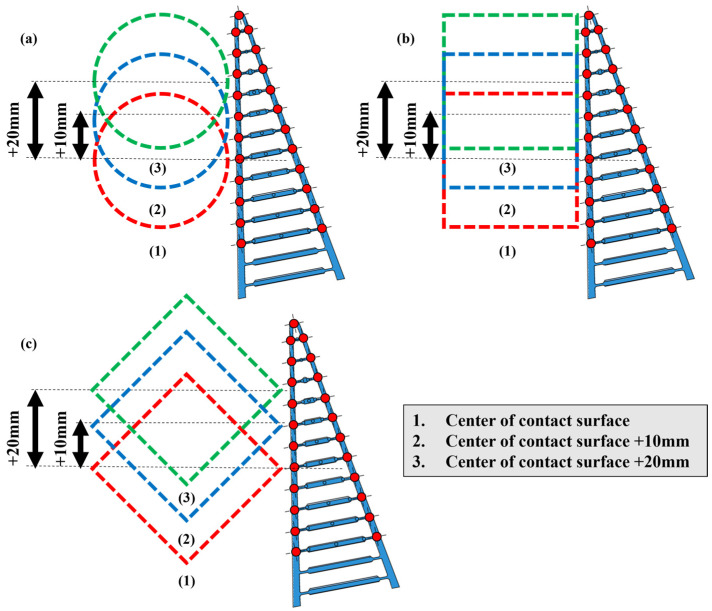
Three different types of grasping position for (**a**) cylinder, (**b**) square (0°), and (**c**) square (45°) objects.

### 2.3. Description of FRSG Specifications

Figure 3a shows the external appearance and dimensions of the FRSG, which opens and closes using a parallel open/close drive system commonly used in industry [27,28]. The parallel open/close mechanism is driven by two motors (Robotis Dynamixel XM540, Robotis Co, Ltd., Seoul, Republic of Korea). T-slots are provided in the left and right drive sections of the FRSG, into which the T-slots of the clips that hold the Fin Ray structure can be inserted. This makes it easy to attach, detach, and replace five different Fin Ray structures. The parallel open/close mechanism was designed in 3D CAD using Autocad’s Fusion360 (2.0.17954 x86_64) and fabricated using an FDM 3D printer (UP300, ABS, Shenzhen Creality 3D Technology Co. Ltd., Shenzhen, China). The amount of opening and closing could be set by motor control.

Five different Fin Ray structures attached to the FRSG have different crossbeam angles, as shown in Figure 3b. These structures were redesigned based on the CAD data of DHAS-GF-120 [8], published by Festo (right side of Figure 3b). The 3D CAD design was carried out using Fusion360, as for the FRSG, and the Fin Ray structures were fabricated using Formlabs’ Form 3 optical 3D printer. The Form 3 3D printer from Formlabs was used to build the Fin Ray structure. This printer is capable of building with UV-curable rubber-like resin (Elastic50A, Yokoito Inc., Kyoto, Japan). Furthermore, a friction test was conducted on a sheet made of Erastic50A on an aluminum plate; the coefficient of static friction was approximately 0.88 and the coefficient of kinetic friction was approximately 0.79. The dimensions of the Fin Ray structure are shown in Figure 3a. In this study, five types of crossbeam (−30°, −15°, 0°, +15°, and +30°) were created and verified to analyze the grasping characteristics of different crossbeams.

Since the gripping characteristics of the FRSG are affected by the crossbeam angle, there is an optimum angle for each of the three shapes of Figure 1c: cylindrical, square (0°), and square (45°). Figure 3 shows the Fin Ray structures (DHAS-GF-120 published by Festo) with a marker used in the previous study and the five Fin Ray structures subject to experimental verification in this study. Each structure is characterized by two features, the first being the contact surface width of the FRSG. It has been confirmed in previous studies that the contact surface and contact force during object grasping and the deformation of the structure are affected. Therefore, the five types of fingers used in this study had a uniform contact surface width to equalize the experiment. The second is the crossbeam angle (−30°, −15°, 0°, +15°, +30°) within the Fin Ray structure. In particular, the five prepared Fin Ray structures have different crossbeam angles from each other. The crossbeam angle is defined as 0° parallel to the bottom beam used to secure the Fin Ray structure to the parallel open/close mechanism.

### 2.4. Experimental Setup

In the first experiment, the participants were asked to grasp a cylindrical object, in the second experiment, a square (0°) object, and in the third experiment, a square (45°) object set up with the square object. Each object shall start the grasping experiment from three different positions: a reference position, a position 10 mm upward from the reference position, and a position 20 mm upward from the reference position, based on the center position of the contact surface of the two Fin Ray structures of the FRSG. Note that the object is fixed to the slide mechanism so that it can move back and forth. There are five types of FRSGs used for grasping with crossbeams of −30, −15, 0, +15, and +30 degrees. Each object and each FRSG are assumed to have constant friction throughout the experiment.

## 3. Experiment 1: Analysis of the Effect of Crossbeam Angle on Grasping Condition

In Experiment 1, five different FRSGs were used to analyze the grasping state of a cylindrical object. The results of cylinder object grasping are shown in four figures; Figure 5 shows the results of the situation in which a particularly stable grasp is achieved among the combinations of each grasping position and FRSG type, Figure 6, Figure 7 and Figure 8 show the contact force and structural deformation for each Fin Ray structure and for each of three different gripping positions (center of contact surface, 10 mm tip from center of contact surface, 20 mm tip from center of contact surface). First, from Figure 6, Figure 7 and Figure 8, the contact force distribution under each condition shows that the center of the contact surface at the grasping position has the widest contact force distribution. It is considered that the pressure is evenly applied to the surface of the grasped object to realize an enveloping grasp. The peak values of the contact force distribution when grasping at the center of the contact surface for the FRSG were 3.3 N (−30°), 3.6 N (−15°), 2.4 N (0°), 4.0 N (15°) and 2.3 N (30°) immediately after the beginning of contact. Furthermore, looking at the movement locus of the marker, it can be confirmed that it deforms most along the shape of the circle when grasped at the center of the contact surface. On the other hand, the position of the cylindrical object during grasping and the position where the object is released after grasping have different characteristics (Table 1, Table 2 and Table 3). In the FRSG (−30°) and FRSG (+30°), when the object is grasped at the center of the contact surface, it is pushed farther out than the starting point of the grasp. In other grasping positions, it tends to be pushed farther out than in the FRSG (−15°), FRSG (0°), and FRSG (+15°). In terms of contact force distribution, the FRSG (−15°), FRSG (0°), and FRSG (+15°) are distributed over a wide range, while FRSG (−30°) and FRSG (+30°) are distributed over a narrow range, which cannot be considered as a wrapping grasp. In the FRSG (−30°) and FRSG (+30°), when the object is grasped at the center of the contact surface, it is pushed farther out than the starting point of the grasp. In other grasping positions, it tends to be pushed farther out than in the FRSG (−15°), FRSG (0°), and FRSG (+15°). In terms of contact force distribution, the FRSG (−15°), FRSG (0°), and FRSG (+15°) are distributed over a wide range. However, the FRSG (−30°) and FRSG (+30°) are distributed over a narrow range, which cannot be considered as a wrapping grasp. This may be due to the influence of the crossbeam angle. Therefore, we discussed the physical properties of each FRSG during grasping: from Figure 5a,c, the FRSGs (−30° and −15°) have a passive deformation function where the contact surface tends to wrap the grasped object within the contact. However, as the crossbeam angle increases in the negative direction, contact between the crossbeams is more likely to occur. In other words, the rate of entrapment increases, but the allowable range of contact deformation decreases. However, the contact between crossbeams causes buckling at the back of the FRSG, generating an S-shape. This prevents the Fin Ray structure from spreading to the outer left and right sides of the FRSG. As a result, the FRSG (−30°) results in a grasping state with a narrow contact area, as shown in Figure 5a. Furthermore, the outward deformation of the FRSG due to the S-shape deformation slides on the contact surface of the grasped object and is pushed forward. In contrast, the FRSG (+30°, +15°) is designed to suppress the crossbeam-to-crossbeam contact that occurs when it contacts the cylinder, as shown in Figure 5b. This allows the raised area near the base of the contact surface of the FRSG to be seen. This raised shape also pushes objects forward with the FRSG. On the other hand, the ideal result can be shown as the FRSGs (0°, 15°) shown in Figure 5c,d. Within these Fin Ray structures, no contact occurs between the crossbeams, the contact force distribution range is widest. It shows the enveloping grasp. And the FRSG deforms along the grasping contact surface, even near the tip of the FRSG; the FRSG is grasped without sliding due to friction from the start of the FRSG grasp until it is closed. How-ever, when the FRSG is opened, it opens from the tip of the FRSG. And it caused slippage via a decrease in the contact area. This prevents the FRSG from returning to its initial position, which is a problem. These misalignments of the object position after grasping need to be considered in the operation of the gripper.

In this experiment, the effect of the crossbeam angle of the FRSG on cylindrical object grasping was verified. The more negative the crossbeam angle is, the more the left–right deformation caused by contact between the crossbeams is suppressed. In addition, the more positive the crossbeam angle is, the more the raised buckling phenomenon at the base of the contact surface is confirmed. Furthermore, it was found that wrapping grasping was best achieved when the crossbeam angle was near 0°. This means that the crossbeam angle has a significant influence on the grasping condition. Countermeasures against object misalignment on the field mounting surface of the FRSG include (1) introducing a position sensor to detect the object position, (2) increasing the friction of the contact surface, and (3) assuming and controlling an error when releasing the object from the FRSG.

**Figure 5 sensors-23-09827-f005:**
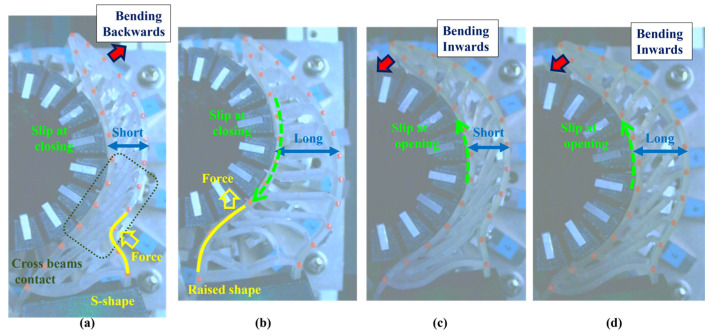
FRPG appearance in cylinder grasp at reference position. (**a**) Crossbeam angle is −30°; (**b**) Crossbeam angle is 30°; (**c**) Crossbeam angle is −15°; (**d**) Crossbeam angle is 0°.

**Table 1 sensors-23-09827-t001:** Object positions from the center of the contact surface for each FRSG operation.

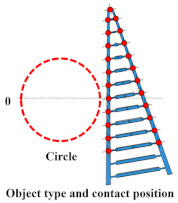		**Handling Start** **[mm]**	**Handling** **[mm]**	**Handling End** **[mm]**
	−30°	0	+9.7 (± 5.1)	+10.8 (±5.3)
	−15°	0	−9.0 (±0.2)	−5.5 (±1.2)
	0°	0	−10.7 (±0.3)	−4.8 (±0.3)
	+15°	0	−11.5 (±0.2)	−1.7 (±0.5)
	+30°	0	+10.2 (±1.2)	+13.8 (±0.8)

**Figure 6 sensors-23-09827-f006:**
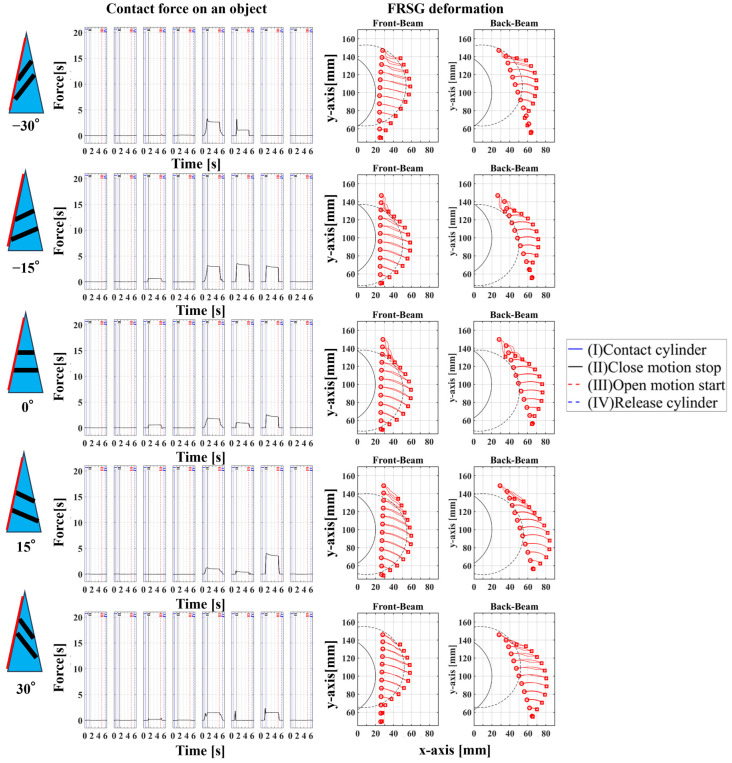
Contact force distribution and structural deformation during object grasping (object: cylinder, grasping position: center).

**Table 2 sensors-23-09827-t002:** Object positions from the center of the contact surface for each FRSG operation.

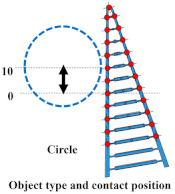		**Handling Start** **[mm]**	**Handling** **[mm]**	**Handling End** **[mm]**
	−30°	10	−2.2 (±0.2)	−0.4 (±0.1)
	−15°	10	−8.9 (±0.2)	−4.6 (±0.2)
	0°	10	−8.1 (±0.1)	−5.2 (±0.3)
	+15°	10	−9.9 (±0.1)	−4.9 (±0.5)
	+30°	10	−0.0 (±0.1)	+3.8 (±0.2)

**Figure 7 sensors-23-09827-f007:**
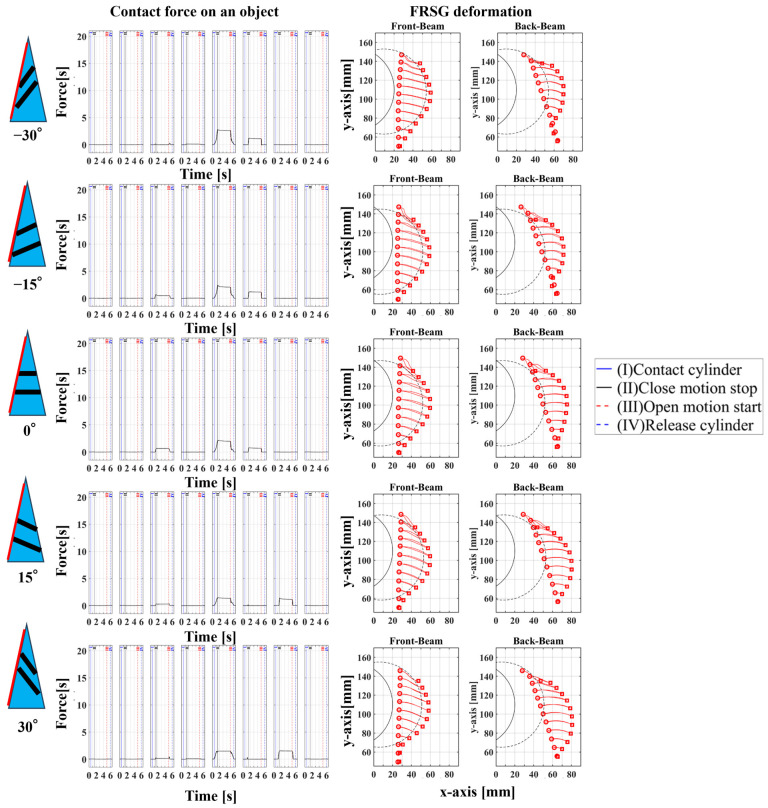
Contact force distribution and structural deformation during object grasping (object: cylinder, grasping position: 10 mm upward from the center).

**Table 3 sensors-23-09827-t003:** Object positions from the center of the contact surface for each FRSG operation.

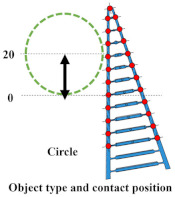		**Handling Start** **[mm]**	**Handling** **[mm]**	**Handling End** **[mm]**
	−30°	20	−3.1 (± 0.3)	−3.0 (±0.3)
	−15°	20	−7.2 (±0.1)	−6.4 (±1.2)
	0°	20	−6.1 (±0.1)	−4.9 (±0.2)
	+15°	20	−7.6 (±0.1)	−7.5 (±0.2)
	+30°	20	−3.2 (±1.2)	−2.9 (±0.8)

**Figure 8 sensors-23-09827-f008:**
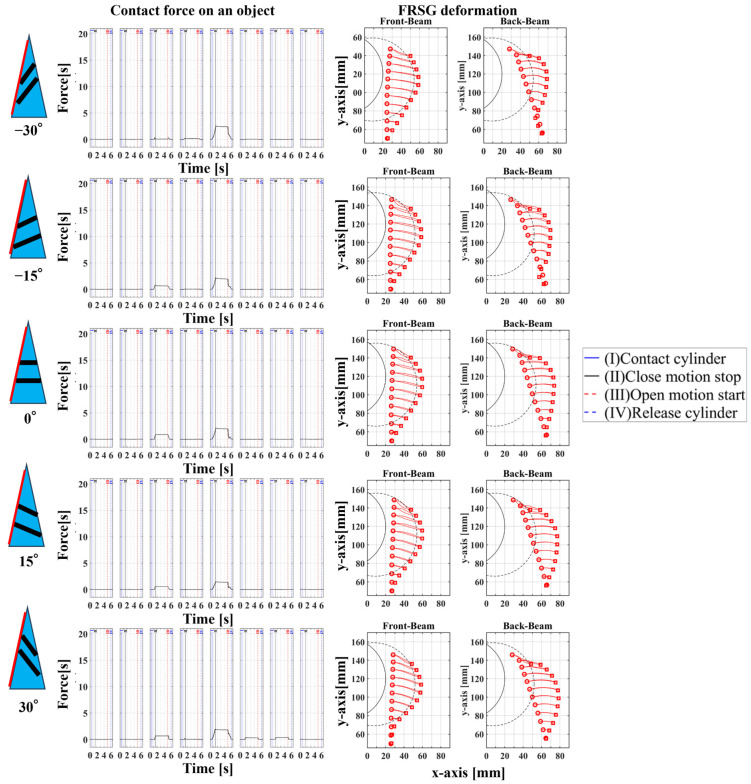
Contact force distribution and structural deformation during object grasping (object: cylinder, grasping position: 20 mm upward from the center).

## 4. Experiment 2: Analysis of Grasping State When Grasping an Angular Object Shape Like a Square

In Experiment 2, we analyzed the crossbeam angles that can realize stable grasping based on the grasping conditions of the corners of the FRSG in a square grasp, which is a common grasping object shape.

### 4.1. Grasping Analysis of Five Types of FRSG for a Square Object (0 Degrees)

Figure 9 shows the results of particularly stable grasping for each combination of grasping position and FRSG type. Figure 10, Figure 11 and Figure 12 show the contact force and structural deformation for each Fin Ray structure and three different grasping positions (center of the contact surface, 10 mm from the center of the contact surface to the tip, 20 mm from the center of the contact surface to the tip). First, the contact force distributions in Figure 10, Figure 11 and Figure 12 confirm that two or three points of contact occur regardless of the grasping position. This indicates that contact is made only at the lower and upper corners of the square in Figure 9. The peak values of the contact force distribution are 20.8 N for the FRSG (0°) for grasping at the center of the contact surface, 14.1 N for the FRSG (−15°) for grasping at 10 mm from the center to the tip, and 4.6 N for the FRSG (−15°) for grasping at 20 mm from the center to the tip. Next, the marker trajectory shows that the grasped object is pushed forward by the FRSG when the crossbeam angle is lowered from the FRSG (−15°). In the case of the FRSG (−30°), the lower corner of the square begins to slide upward on the FRSG contact surface, as shown in Figure 9a. The phenomenon occurs when the object leaves the FRSG graspable area (Table 4, Table 5 and Table 6). In the FRSG (+15°, +30°), as shown in Figure 9b, the effect of suppression of contact between crossbeams can be confirmed when in contact with the lower corner of the square body. Conversely, buckling of the contact surface near the base of the FRSG occurs. Excessive back beam bending pushes the object forward, causing the object to move away from the FRSG grasping area. Even when the object position is +20 mm from the center of the contact surface, the tip of the FRSG goes around the upper corner of the square, making it difficult to get inside the right side of the square. This is one of the reasons why it is pushed forward. Therefore, we consider it a prerequisite for stable grasping that the tip of the FRSG is well above the upper angle of the square body and that it holds the corner. As a consideration for square (0°) grasping, the ideal grasping position for the FRSG (+15°) was at the center of the contact surface where the object was grasped from. However, instead of closing the FRSG to the end (causing breakage), as in Figure 9c, the lower corner of the square body should contact the FRSG with no contact between the crossbeams, as in Figure 9d. Furthermore, the tip of the FRSG should show a wrapping motion and contact the upper corner of the square body. These two conditions are the prerequisites for an ideal pincer grasp. The flow to determine if the FRSG is gripped stably is to measure the contact force (6ch, 7ch) between the area near the base of the FRSG and the lower corner of the square body during the closing movement of the FRSG. Before the value becomes excessively high, the presence or absence of contact force (2ch, 3ch) near the tip of the FRSG with the upper angle of the square body is checked, and if there is enough contact force, the closing operation is stopped.

**Figure 9 sensors-23-09827-f009:**
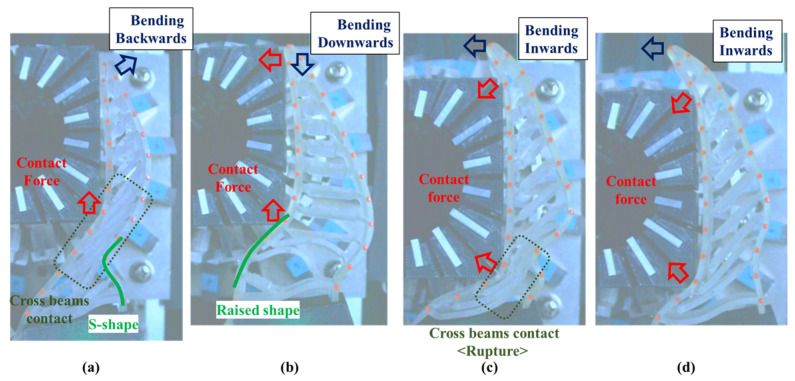
FRPG appearance in square (0°) grasp at reference position. (**a**) Crossbeam angle is −30°; (**b**) Crossbeam angle is 30°; (**c**) Crossbeam angle is −15°; (**d**) Crossbeam angle is 0°.

In this verification, the effect of the crossbeam angle on the stable grasping of a square (0°) was confirmed by analyzing the grasping condition of the corner of the square (0°). The guidelines for judging the completion of stable grasping were also clarified.

**Table 4 sensors-23-09827-t004:** Object positions from the center of the contact surface for each FRSG operation.

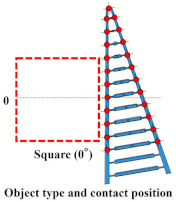		**Handling Start** **[mm]**	**Handling** **[mm]**	**Handling End** **[mm]**
	−30°	0	+29.1 (±0.0)	+29.0 (±0.0)
	−15°	0	−2.1 (±0.7)	−4.0 (±1.0)
	0°	0	−1.9 (±0.1)	+7.4 (±2.4)
	+15°	0	−1.4 (±0.8)	+8.1 (±0.9)
	+30°	0	+23.1 (±0.6)	+29.1 (±0.0)

**Figure 10 sensors-23-09827-f010:**
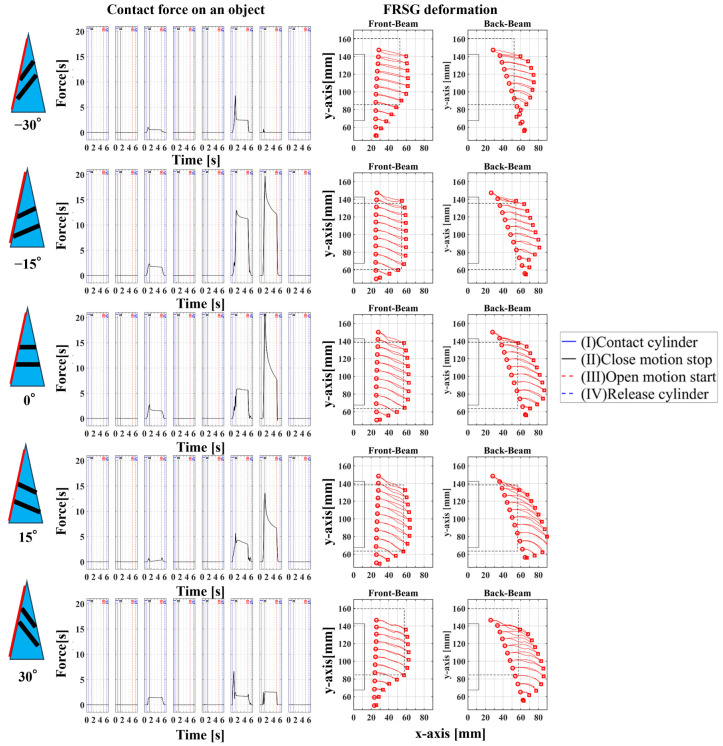
Contact force distribution and structural deformation during object grasping (object: square (0°), grasping position: center).

**Table 5 sensors-23-09827-t005:** Object positions from the center of the contact surface for each FRSG operation.

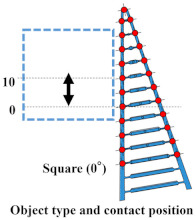		**Handling Start** **[mm]**	**Handling** **[mm]**	**Handling End** **[mm]**
	−30°	10	+19.0 (±0.0)	+19.0 (±0.0)
	−15°	10	−6.7 (±0.1)	−3.4 (±0.3)
	0°	10	−9.9 (±0.1)	−4.7 (±0.8)
	+15°	10	−10.6 (±2.0)	+1.1 (±2.0)
	+30°	10	+10.0 (±1.8)	+14.5 (±2.2)

**Figure 11 sensors-23-09827-f011:**
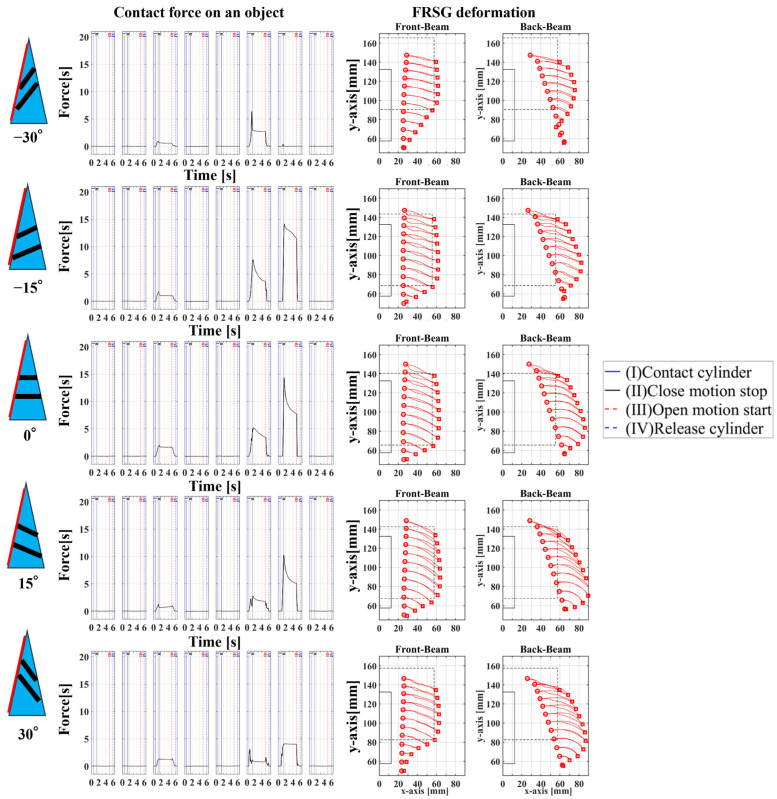
Contact force distribution and structural deformation during object grasping (object: square (0°), grasping position: 10 mm upward from the center).

**Table 6 sensors-23-09827-t006:** Object positions from the center of the contact surface for each FRSG operation.

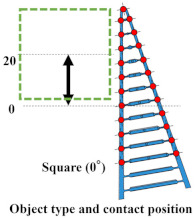		**Handling Start** **[mm]**	**Handling** **[mm]**	**Handling End** **[mm]**
	−30°	20	9.0 (±0.0)	9.0 (±0.0)
	−15°	20	−8.9 (±0.4)	−4.0 (±7.0)
	0°	20	−12.2 (±0.1)	−7.5 (±0.3)
	+15°	20	−13.5 (±0.1)	−5.4 (±0.6)
	+30°	20	+0.6 (±0.4)	+6.5 (±1.4)

**Figure 12 sensors-23-09827-f012:**
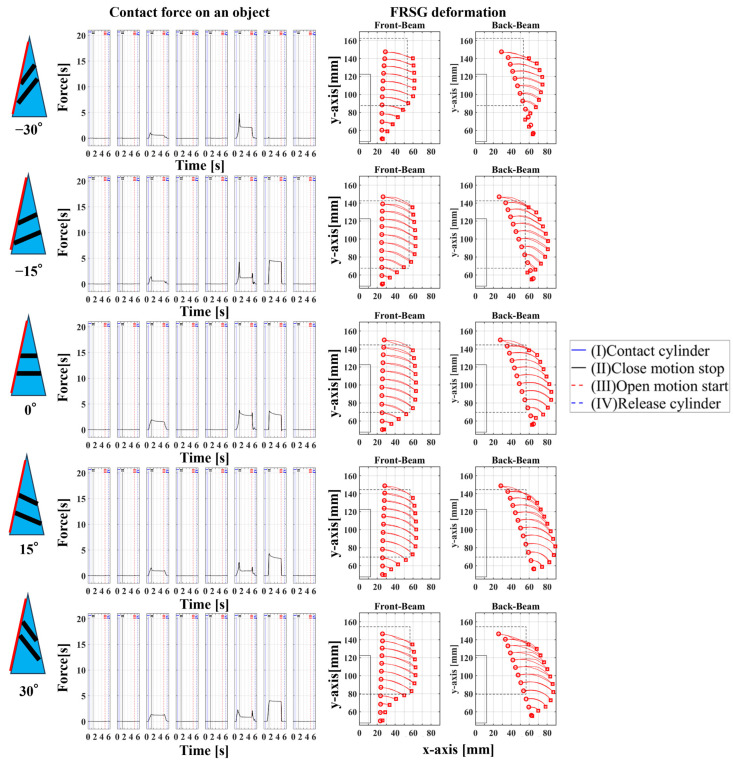
Contact force distribution and structural deformation during object grasping (object: square (0°), grasping position: 20 mm upward from the center).

### 4.2. Grasping Analysis of Five Types of FRSG for a Square Object (45 Degrees)

Next, this experiment analyzes the grasping conditions when the square is rotated 45° and verifies the optimal shape and grasping strategy for stable grasping of the corner. Figure 13 shows the Fin Ray structure deformation during a square (45°) grasp. Figure 14, Figure 15 and Figure 16 show the results for each starting position of the grasp. The contact force distributions in Figure 14, Figure 15 and Figure 16 show that the peaks appear at almost the same locations. The maximum force at the FRSG (−15°) is 7.3 N, 5.8 N, and 4.2 N for the case of grasping at the center of the contact surface, at 10 mm from the center of the contact surface to the tip, and at 20 mm from the center of the contact surface to the tip, respectively. A feature common to all five types of FRSG is that when the contact surface is brought into contact with an object, it tends to retract to the root side of the FRSG due to entrainment forces (Table 7, Table 8 and Table 9). In the case of the FRSG (−30°, −15°), the tip of the FRSG tends to deform in such a way that it wraps around the object. When the object contacts near the base of the FRSG, the distance between the crossbeams near the object contact position immediately disappears, as shown in Figure 13a. It is thought that the functionality of the passive motion is lost due to the contact and integration of the crossbeams. In addition, the back beam buckles into an S-shape, and the FRSG bends outward when excessively bent. This tended to prevent the tip of the FRSG from contacting the object. In contrast, in the case of the FRSG (+15°, +30°), as shown in Figure 13b, the contact between the crossbeams near the object contact position was suppressed even when in contact with the object. On the contrary, buckling was observed near the root, where the contact surface was raised up against the object, and excessive bending pushed the object forward. In moderate cases, the forward push was effective in gripping with the tip of the FRSG. Conversely, when it was excessive, the tip of the FRSG was sometimes forced to warp, making grasping impossible. As a result, it was found that the FRSG (0°) had both functions in moderation, and that it was ideal to achieve either a two-point contact between the tip and the center of the contact surface (Figure 13c), or a three-point contact between the tip, the center, and near the bottom of the contact surface (Figure 13d).

As a final guess, the ideal FRSG for grasping a square (45°) object does not have all crossbeams in the structure at the same angle. It is effective to reduce contact between crossbeams below the object contact point by setting the crossbeam angle to +15° or +30°. The raised structure below the contact surface is used to create a functional structure. The most ideal situation would be to set the crossbeam angle at −15° or −30° above the object contact point to create a wraparound grasping function. To clarify the stability conditions of angled object grasping by the FRSG, we performed grasping on a square 0° configuration and a square 45° configuration of objects and verified the results with multiple FRSGgeometries. As a result, it was clarified that the combination of different crossbeam angles at the contact positions of the corners can generate a contact or non-contact condition between the crossbeams, which results in more stable grasping.

**Figure 13 sensors-23-09827-f013:**
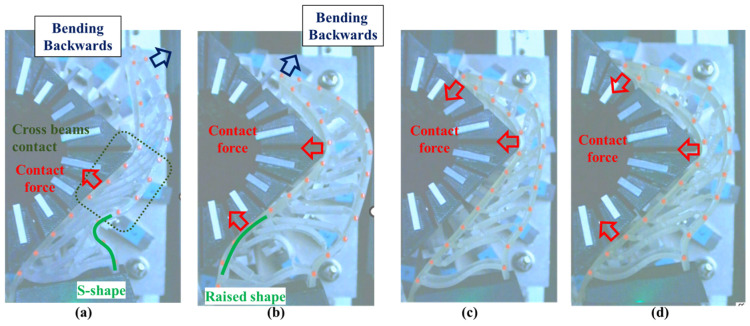
FRPG appearance in square (45°) grasp at reference position. (**a**) Crossbeam angle is −30°; (**b**) Crossbeam angle is 30°; (**c**) Crossbeam angle is −15° at +20 mm from center of contact surface; (**d**) Crossbeam angle is 0° at +10 mm from center of contact surface.

**Table 7 sensors-23-09827-t007:** Object positions from the center of the contact surface for each FRSG operation.

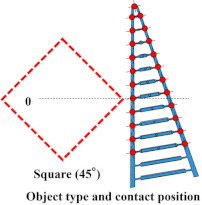		**Handling Start** **[mm]**	**Handling** **[mm]**	**Handling End** **[mm]**
	−30°	0	−0.6 (±0.9)	4.1 (±0.7)
	−15°	0	−7.1 (±0.1)	−2.8 (±0.2)
	0°	0	−8.2 (±0.1)	−4.2 (±0.5)
	+15°	0	−3.5 (±1.9)	1.5 (±2.6)
	+30°	0	+7.5 (±0.5)	+9.1 (±2.4)

**Figure 14 sensors-23-09827-f014:**
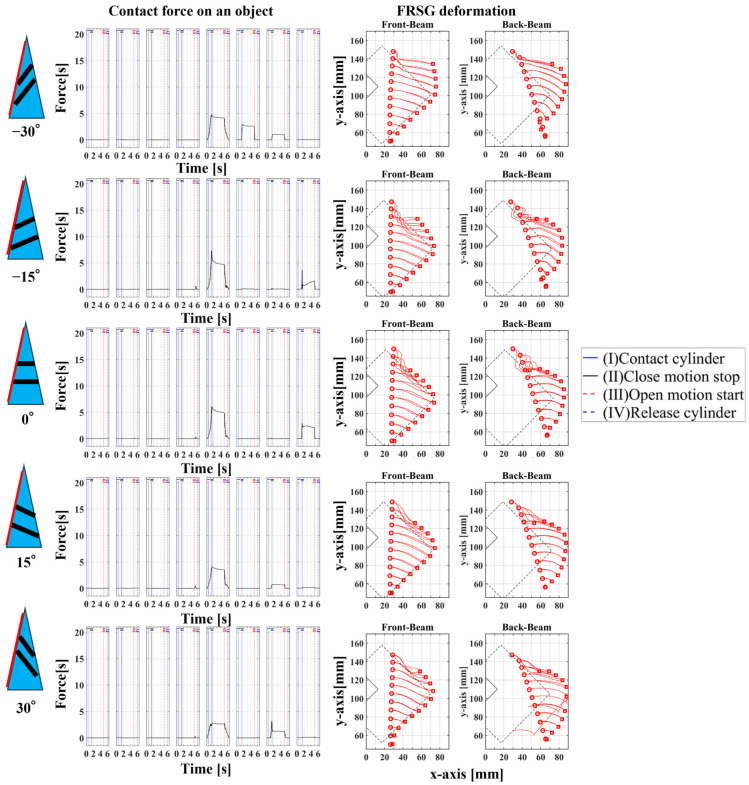
Contact force distribution and structural deformation during object grasping (object: square (45°), grasping position: reference position).

**Table 8 sensors-23-09827-t008:** Object positions from the center of the contact surface for each FRSG operation.

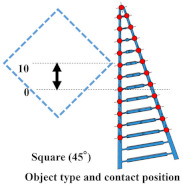		**Handling Start** **[mm]**	**Handling** **[mm]**	**Handling End** **[mm]**
	−30°	10	−3.8 (±0.6)	4.1 (±0.7)
	−15°	10	−11.7 (±0.6)	−7.4 (±0.4)
	0°	10	−12.7 (±0.5)	−8.5 (±0.7)
	+15°	10	−12.8 (±0.5)	−5.4 (±0.8)
	+30°	10	−0.6 (±0.2)	+4.1 (±1.8)

**Figure 15 sensors-23-09827-f015:**
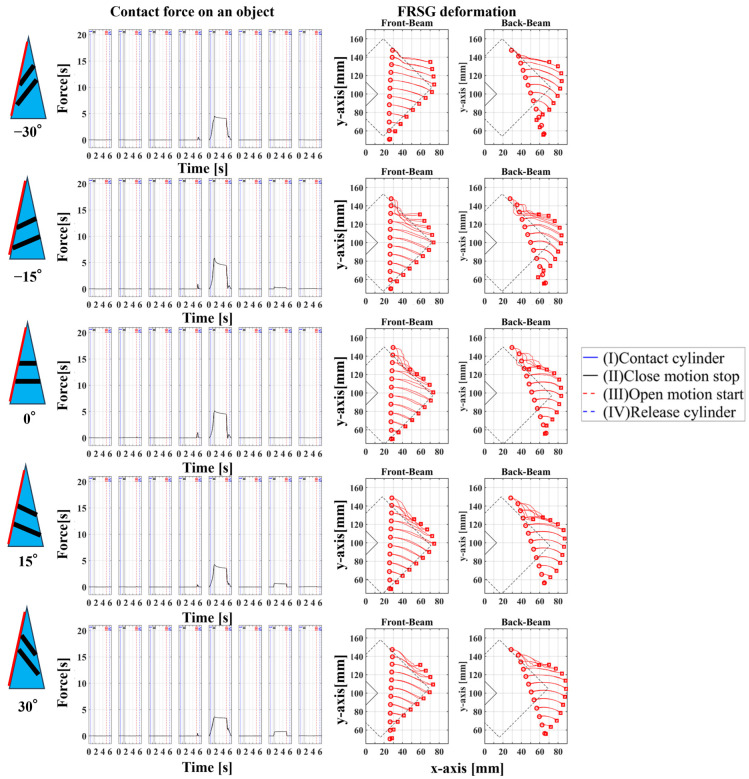
Contact force distribution and structural deformation during object grasping (object: square (45°), grasping position: 10 mm upward from the center).

**Table 9 sensors-23-09827-t009:** Object positions from the center of the contact surface for each FRSG operation.

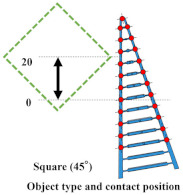		**Handling Start** **[mm]**	**Handling** **[mm]**	**Handling End** **[mm]**
	−30°	10	−3.8 (±0.6)	4.1 (±0.7)
	−15°	10	−11.7 (±0.6)	−7.4 (±0.4)
	0°	10	−12.7 (±0.5)	−8.5 (±0.7)
	+15°	10	−12.8 (±0.5)	−5.4 (±0.8)
	+30°	10	−0.6 (±0.2)	+4.1 (±1.8)

**Figure 16 sensors-23-09827-f016:**
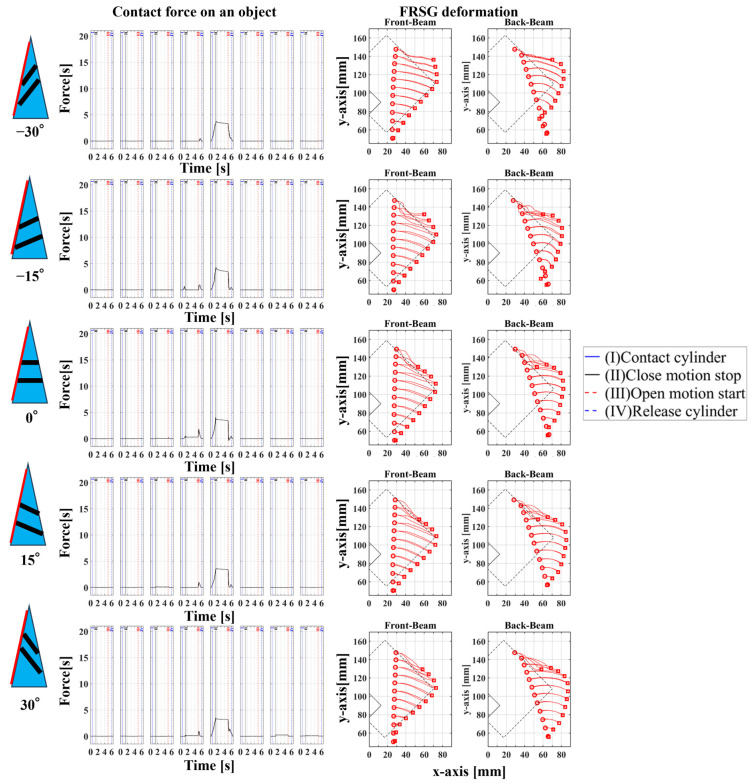
Contact force distribution and structural deformation during object grasping (object: square (45°), grasping position: 20 mm upward from the center).

## 5. Conclusions

In the study of Fin-Ray-type soft grippers (FRSGs), researchers have developed a wide variety of FRSGs and evaluated their operation. In many studies, buckling and slippage were difficult to describe by FEM analysis, and contact analysis of angled objects was not consistent with the actual motion of the actual device. On the other hand, it has been reported that the grasping characteristics of FRSGs vary depending on the crossbeam angle, and it is necessary to verify this in the actual machine. Therefore, in this study, we analyzed the influence of the crossbeam angle on the grasping characteristics and the grasping state of an angled object to establish a guideline for determining the FRSG structure to suit the task. The analysis was carried out using a discarded body developed in a previous study by our research team, which is a combination of a contact force measuring device (type 16/16) with a force sensor built into the grasping object and a structural deformation measuring device using camera image processing. Five different FRSGs (−30°, −15°, 0°, +15°, and +30°) were prepared, differing only in crossbeam angle. Using these fingers, grasping experiments were conducted with three types of grasping objects: a cylinder, which is the basic shape of the object to be grasped, a square 0° configuration and a square 45° configuration, which is assumed to be used in daily life. As a result, it was confirmed that the FRSG (0°, −15°) performs a particularly extensive enveloping grasp for cylinders. However, the initial position of the grasped object tended to differ from the position after the end of grasping, indicating that countermeasures were necessary. Next, for the square (0°), the best results were obtained for the FRSG (+15°). These results indicate that for stable grasping of a square object, one Fin Ray structure must make two points of contact with the corners of the square object. Therefore, we also found that there are two important grasping behaviors: the first is that the FRSG raised buckling structure can contact the lower corners of the square body and maintain the crossbeam spacing. The second is that the tip of the FRSG is at the top of the square body and can create a wraparound shape change. Finally, the best results were obtained when the FRSG (0°) was square (45°): above the contact position between the contact surface of the FRSG and the corner of the square body, a wrapping grasp shape change is required and a crossbeam angle of −30° was used. Furthermore, below the contact position, it is appropriate to configure the crossbeam angle at +30° to prevent crossbeam-to-crossbeam contact. It was estimated that this would enable stable square wraparound grasping.

## Figures and Tables

**Figure 1 sensors-23-09827-f001:**
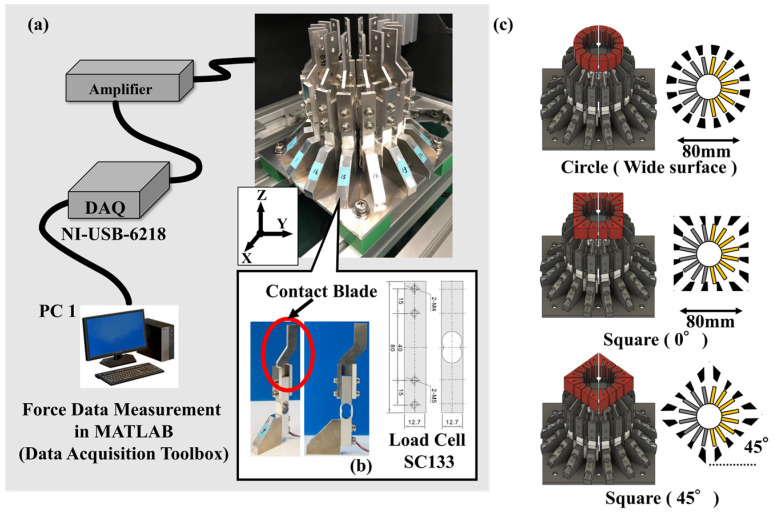
Contact force distribution measurement device with multiple force sensors built into the gripping body. (**a**) Conceptual diagram of the device; (**b**) Force sensor and contact part mounting; (**c**) Shape of the grasped object (cylinder, square 0°, square 45°).

## Data Availability

The data used in this work are available upon request to the corresponding author.

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
