# Peer review of "Analysis of Contact Force and Shape Change on Grasping a Square Object Using an Actual Fin Ray Soft Gripper"

_sensors, 2023, doi:10.3390/s23249827_

Round 1

Reviewer 1 Report

Comments and Suggestions for Authors

The submitted study investigates contact force and shape change on grasping objects using a manufactured FinRay-type soft gripper. The authors need to address the below points.

a) Indicate the difference between FRSGs and the other soft robotics grippers below, in the Introduction part of the article.

-       Dilibal, S, Sahin, H, Danquah, J.O. et al. (2021) Additively Manufactured Custom Soft Gripper with Embedded Soft Force Sensors for an Industrial Robot. Int. J. Precis. Eng. Manuf. 22, 709–718 https://doi.org/10.1007/s12541-021-00479-0

      Sun, T., Chen, Y., Han, T., Jiao, C., et al. (2020) A soft gripper with variable stiffness inspired by pangolin scales, toothed pneumatic actuator, and autonomous controller. Robot CIM-INT Manufacture, 61, 1–12 doi.org/10.1016/j.rcim.2019.101848

b) What is the maximum contact force obtained from the manufactured FinRay-type soft grippers?

c) What is the expected friction based on the type of material that is in contact.

d) Reorganize and clarify the Conclusion part of the article via comparing all of the received results.

Author Response

Thank you for your time。
Please read through the documents attached file.

Reviewer 2 Report

Comments and Suggestions for Authors

This work proposed a force measurement system for validating the performance of the soft gripper under different manipulation tasks. The system setup and mechanism are introduced, experiments are conducted for demonstration. Overall the quality is good, and it should meet the acceptance requirement after comments below are properly addressed.

·      The abstract should be simplified, currently it is more like conclusion section.

·      Page 3 line 102, the previous research conducted by the authors should be cited here.

·      A summary of contribution should be provided at the end of section 1 to demonstrate the novelty of this work against the authors’ previous work.

·      The writing should be improved, a lot of the context can be simplified and polished to show the focus.

Comments on the Quality of English Language

the writing can be improved for easier understanding.

Author Response

Thank you for your time.
Please read through the documents attached file.

Round 2

Reviewer 1 Report

Comments and Suggestions for Authors

I examined the revised version of the paper. The revised paper can be published in Sensors.